# A Practical Guide to Injection Therapy in Hand Tendinopathies: A Systematic Review of Randomized Controlled Trials

**DOI:** 10.3390/jfmk9030146

**Published:** 2024-08-26

**Authors:** Antonio Frizziero, Nicola Maffulli, Chiara Saglietti, Eugenio Sarti, Davide Bigliardi, Cosimo Costantino, Andrea Demeco

**Affiliations:** 1ASST “Gaetano Pini” CTO, 20122 Milano, Italy; 2Faculty of Medicine and Psychology, University La Sapienza, 49911 Rome, Italy; n.maffulli@qmul.ac.uk; 3Department of Medicine and Surgery, University of Parma, 43126 Parma, Italy; chiara.saglietti@unipr.it (C.S.); eugenio.sarti@unipr.it (E.S.); davide.bigliardi@unipr.it (D.B.); cosimo.costantino@unipr.it (C.C.); andrea.demeco@unipr.it (A.D.)

**Keywords:** rehabilitation, hyaluronic acid, pain treatment, de Quervain’s tenosynovitis, trigger finger

## Abstract

Hand tendinopathies represent a pathological condition associated with significant disability. However, due to this high heterogeneity of the treatments and their efficacy, there is still a lack of consensus on the infiltrative therapy of the hand. This systematic review aimed to investigate the efficacy of injection techniques in the treatment of pain related to the main hand tendinopathies. We searched online medical databases (PubMed, Pedro, Cochrane Library, Scopus, and WoS). Only RCTs published in the last 10 years (up to 5 August 2024), written in English, and related to infiltrative treatment in wrist and hand tendinopathies were evaluated. The risk of bias in RCTs was assessed with Version 2 of the Cochrane Risk of Bias tool for randomized trials (RoB 2). Out of 641 articles identified, 23 were included in the final synthesis: 14 RCTs on trigger finger, and 9 RCTs on de Quervain’s tenosynovitis. The present systematic review showed that infiltrative therapy of trigger finger and de Quervain’s tenosynovitis constitutes a fundamental element in the treatment of these pathological conditions, in terms of pain reduction and improvement in the functionality of the hand.

## 1. Introduction

Tendons connect muscles to bones, allowing the muscular system to perform its contractile functions. Histologically, tendons are composed of a cellular population and highly organized connective tissue. The cellular part includes tenocytes, tenoblasts, and chondrocytes; the connective tissue is composed of type I collagen fibers and an extracellular matrix (ECM) that contains proteoglycans and elastin. Type I collagen is organized in a specific three-dimensional architecture of fibrils, fibers, and bundles [1]. Moreover, there are three membranous structures called endotenon, epitenon, and paratenon: the endotenon surrounds each tendon fiber, and the epitenon and the paratenon surround the entire tendon structure [2].

The term “tendinopathy” includes different pathological conditions of the tendon, with different histological alterations, but commonly characterized by pain, tenderness, and reduced function [3]. Although the term “tendinitis” is still commonly used, the most recent studies that analyze the histological aspect of tendons affected by a chronic disorder have shown that there is not any feature typical of inflammatory processes. In these cases, the term “tendinopathies” is the most correct because the histological aspects commonly found are loss of normal fibrillar architecture, degeneration of the collagen component, general thickening of the tendon, and alteration of the normal healing process [2,4].

During the evaluation of tendinopathy, it is important to identify the portion of the tendon affected by pathological alterations, classifying the tendinopathy as insertional and non-insertional forms. In the first one, the pathological process affects the osteo-tendinous junction (enthesis); conversely, in non-insertional forms, pathological alterations involve the central portion of the tendon [5].

In general, the highest incidence of tendinopathies occurs between the second and the sixth decade [6,7].

Hand tendinopathies represent a pathological condition associated with significant disability, especially in the working population [8,9,10].

Numerous risk factors have been identified for the development of tendinopathies. Risk factors can be divided into two main categories: intrinsic and extrinsic factors. The first group includes genetic factors, age, the influence of sex hormones, endocrine disorders such as thyroid function alterations, metabolic disorders such as diabetes mellitus, obesity, inflammatory conditions, limited or excessive joint mobility, and deficits in neuromuscular control. On the other hand, the second group includes smoking, drugs (fluoroquinolones, corticosteroids, and statins), and conditions with excessive tendon strain (“overuse”) during work or sport activities or, on the other hand, insufficient stimulation (“underuse”) such as in situations of a sedentary lifestyle or excessively prolonged functional rest [6]. In particular, athletes can develop tendinopathies in conditions of excessive functional demand associated with repetitive movements that stress the tendon beyond its physiological resistance capacity [11].

Tendinopathies represent a common pathological condition and the most frequently affected sites of the upper limb are the shoulder (rotator cuff tendons), the elbow (common extensor tendon) [6,12,13], and the hand (flexor and extensor tendons of the fingers, tendons of abductor pollicis longus and extensor pollicis brevis, extensor pollicis longus, extensor carpi ulnaris, flexor carpi radialis, and flexor carpi ulnaris) [14].

The tendons located in the anatomical district of the wrist and hand are structurally surrounded by a synovial sheath, which allows for the tendon to slide correctly inside it, thanks to the presence of a minimal amount of synovial fluid, also known as “sliding tendons”. In case of overuse, there is an altered size ratio between the tendon and its surrounding sheath or pulley, known as stenosing tenosynovitis, resulting in discomfort or pain [15].

In particular, it is defined as “De Quervain’s tenosynovitis”, the stenosing tenosynovitis involving the abductor pollicis longus and extensor pollicis brevis tendons as they pass within the first dorsal compartment of the extensor retinaculum at the wrist [16]; and “trigger finger”, the stenosing tenosynovitis that affects the superficial and deep flexor tendons of the fingers, as well as the long flexor tendon of the thumb, and the pulleys through which they pass. Typically, the A1 pulley, located at the metacarpophalangeal joint, is involved [17,18].

The risk factors for the development of de Quervain’s tenosynovitis are functional overload related to work or sport activities that involve a repetitive movement of ulnar deviation of the wrist (using work tools such as a hammer, racquet sports, skiing, rowing, golf, volleyball, and bowling), chronic inflammatory diseases, and local traumas. The subjects most affected are women and the dominant hand is most frequently involved [14,16,19]. Trigger finger also predominantly affects women and exhibits a higher prevalence on the dominant hand. The primary etiopathogenetic factors encompass metabolic and rheumatological pathologies, local traumas, and functional overuse. Specifically, the incidence of trigger finger is elevated in individuals engaged in repetitive finger movements or prolonged forceful grips during occupational or sports activities [20,21].

Clinically, hand and wrist tendinopathies manifest with symptoms such as pain during both passive and active movement, localized tenderness to digital pressure, swelling, and reduced functionality of the affected hand and wrist region [14,15,19].

Specifically, de Quervain’s tenosynovitis is characterized by pain and tenderness localized on the radial side of the wrist and exacerbated during specific movements such as grasping objects involving ulnar deviation of the wrist. Symptoms may include skin redness and local swelling over the radial styloid process. The primary semiotic test is the Finkelstein test, in which the examiner passively induces ulnar deviation of the patient’s wrist while the thumb is gripped by the other fingers. The test is considered diagnostic if it elicits the typical pain [14,15,19].

In trigger fingers, clinical findings include hand pain triggered both by flexion and extension movements of the metacarpophalangeal and proximal interphalangeal joints of the affected finger, as well as tenderness upon the palpation of pulley A1. Moreover, patients often report a sensation of “snapping” during flexion–extension movements of the finger affected by the condition. As the condition worsens, the affected finger may become stuck in a flexed or extended position, requiring passive release. In advanced stages, it can lead to non-correctable locking, neither actively nor passively. Additionally, palpation of the palm of the hand, at the site where stenosing tenosynovitis has developed, may reveal a painful subcutaneous nodule with an elastic consistency [14,15,19].

The optimal management of tendinopathies involves a multidisciplinary treatment based on various therapeutic options, although these several methods share the same therapeutic purposes: reducing pain, improving functionality, accelerating healing, and increasing the patient’s quality of life. Treatments are divided into passive modalities, such as pharmacological therapies, infiltrative therapies, physical therapies through the use of shock waves (ESWT) or low-level laser therapy, and active modalities, such as tendon load exercise, patient education, and load management [22]. The multiple modalities can be used individually or in combination (e.g., in synergy with therapeutic exercises) [6].

In this context, therapeutic exercise plays a key role. In particular, eccentric exercise has shown greater therapeutic efficacy through the application of a load on the muscle–tendon system during the lengthening phase. Eccentric exercise has been shown to stimulate the synthesis of collagen fibers with a remodeling of the tendon and has given positive effects in terms of pain reduction. However, the exact number of weekly and daily sessions, sets, and repetitions for each type of exercise has not yet been defined [23,24].

Regarding therapy with non-steroidal anti-inflammatory drugs (NSAIDs), the most recent scientific evidence leads to the conclusion that there is no indication for the use of this class of drugs in chronic tendinopathies due to the absence of an ongoing inflammatory process. Moreover, a negative effect of NSAIDs on tendon regeneration has been highlighted in the case of frequent use of these drugs. Therefore, the use of NSAIDs should be limited to the treatment of acute tenosynovitis or reactive tendinopathy [4,25,26].

Instrumental physical therapies are a valid approach to the integrated treatment of tendinopathies. In particular, those that have shown more evidence in terms of therapeutic efficacy are high- and low-level laser therapy and extracorporeal shockwave therapy.

Low-level laser therapy (LLLT) takes advantage of the properties of laser light or low-power LED. Its power range is between 10 mW and 500 mW, and its wavelength is between red and infrared. LLLT has the ability to penetrate the skin and soft tissues and has the characteristics of modulation of pain, an anti-inflammatory effect, and tissue regeneration. LLLT also does not cause overheating of the skin [27].

High-level laser therapy (HLLT), compared to LLLT, is characterized by high intensity (>0.5 W/cm^2^) and low frequency (10–40 Hz) pulses. An example of HLLT used in musculoskeletal rehabilitation is the Nd:YAG laser which has an average power of 0.3–10 Watts and a high peak power (200–3000 Watts). This type of laser has a high tissue penetration capacity but produces a thermal effect that causes heating in the superficial layers of the skin [28,29].

Extracorporeal shockwave therapy (ESWT) produces high energy pressure waves, which penetrate the tissues and have several properties including neoangiogenesis stimulation and anti-inflammatory effect [30].

Nowadays, there is growing evidence of the role of nutraceuticals in the integrated treatment of tendinopathies. Nutraceuticals are substances derived from plants, microbial agents, and foods. The most studied substances are glucosamine, chondroitin sulfate, vitamin C, collagen I, L-arginine-α-keto-glutarate, curcumin, boswellic acid, methilsulfonilmethane (MSM), and bromelain. These substances perform through different mechanisms: anti-inflammatory effects, reduction in pain, stimulating effects for collagen production, and improvement in the structural organization of the tendon [24,31,32].

Infiltrative therapy plays a key role in tendinopathies employing various substances including corticosteroids, PRP, autologous blood injections, autologous cell injections, and hyaluronic acid. Numerous pieces of evidence support the importance of using an instrumental guide to support the injection as it increases the precision of the therapeutic act [33,34]. In vitro studies have shown that the therapeutic effect of corticosteroids on the tendon and surrounding connective tissues is based on an important anti-inflammatory effect associated with the inhibition of collagen, extracellular matrix molecules, and granulation tissue. However, these effects are only present in the short term. Chronic corticosteroid treatment can result in numerous adverse effects, including tendon rupture [35]. In general, the use of corticosteroids in the treatment of tendinopathies has decreased due to different conflicting results, depending on the body district in which the tendons are affected [6].

Platelet-rich plasma (PRP) is an autologous blood preparation, which is centrifuged isolating a high concentration of platelets, with or without leukocytes. Platelet degranulation releases several factors including TGFβ, PDGF, bFGF, VEGF, IGF1, and EGF, which are involved in different stages of tendon healing; their presence in the tendinopathic environment may play a role in accelerating the healing and regeneration of damaged tendon tissue [36,37].

Finally, recent studies are investigating the therapeutic effect of peritendinous hyaluronic acid (HA) infiltrations in the treatment of tendinopathies with encouraging results. In particular, a positive effect in terms of reducing pain and improving function has been described. From a biological point of view, HA improves tendon viscoelasticity together with a lubricant, protective, and anti-inflammatory effect. In addition, no major side effects related to HA injections were described during these studies [4,38].

However, due to this high heterogeneity of the treatments and their efficacy, there is still a lack of consensus on the infiltrative therapy of hand tendinopathies. Thus, in the present systematic review, we sought to investigate the efficacy of injection techniques in the treatment of pain related to the main tendon pathologies of the hand.

## 2. Materials and Methods

### 2.1. Search Strategy

The present systematic review was conducted by searching online medical databases (PubMed, Pedro, Cochrane Library, Scopus, and WoS). Only randomized controlled clinical trials (RCTs) published in the last 10 years (up to 5 August 2024), written in English, and related to infiltrative treatment in wrist and hand tendinopathies were evaluated.

The studies were selected using the following search string: ((injection) OR (infiltrative treatment) OR (infiltration)) AND ((tendinopathy) OR (tendinopathies) OR (tendinosis) OR (tendon injury) OR (tendon inflammation) OR (tendon degeneration)) AND ((upper limb) OR (wrist) OR (thumb) OR (hand)).

Based on the aforementioned criteria, an initial selection of relevant studies was made after reviewing the title and/or abstract. Subsequently, the full text of the selected articles was reviewed for final inclusion in the review, and the bibliography of the individual selected articles was evaluated. Many studies were excluded after reviewing the full text, while others were retrieved from the bibliography of the selected articles.

### 2.2. Selection of Articles

After removing duplicates, 2 reviewers independently screened all papers for eligibility. In case of disagreement, consultation with a third reviewer allowed for consensus. Articles were considered eligible if they agreed with the items defined by the following PICO model: (P) participants: patients with a diagnosis of trigger finger or de Quervain’s tenosynovitis; (I) intervention: infiltrative injection of corticosteroids and hyaluronic acid; (C) comparator: infiltrative injection of other therapeutic substances (NSAIDs and local anesthetics), rehabilitative techniques (ESWT, acupuncture, or using of a brace), or surgical techniques (percutaneous technique or open surgery); (O) outcome measure: pain intensity (VAS or NRS), disability level (DASH, Michigan Hand Outcome Questionnaire, modified patient global impression of improvement, or patient-rated wrist evaluation), ROM quantification (TAM analysis), strength quantification (dynamometer), specific rating scales or tests for pathology (Quinnell grading system, trigger finger score, or Finkelstein test), US evaluation, and complications following therapy.

### 2.3. Data Extraction

Two reviewers independently extracted data from the included studies using customized data extraction on a Microsoft Excel sheet. In case of disagreement, consensus was achieved with a third reviewer. We extracted the following data: (1) first author, (2) publication year, (3) nationality, (4) population and number of patients included, (5) injection technique, (6) experimental group (type of drug and dosage), (7) control group (type of drug and dosage, rehabilitation therapy or surgery), (8) follow-up, (9) outcomes, and (10) conclusions and main findings.

### 2.4. Quality Assessment

The selected studies were synthesized by describing the extracted data. The risk of bias in the RCTs was assessed by 2 reviewers using Version 2 of the Cochrane Risk of Bias tool for randomized trials (RoB 2). Any disagreement was discussed with a third reviewer.

## 3. Results

The initial search yielded 641 articles, of which 29 were identified as potentially relevant after conducting the initial title/abstract screening. Six (6) articles were excluded after reading the entire text due to methodological issues (e.g., non-RCT study design, inclusion of patients with complete tendon ruptures, tendinopathies secondary to rheumatological/neurological/diabetes conditions, etc.). Finally, 23 articles were included, of which there are 14 RCTs on trigger finger and 9 RCTs on de Quervain’s tenosynovitis.

The results of the search and the corresponding selection process are illustrated in the PRISMA flowchart below (Figure 1).

### 3.1. Trigger Finger

Fourteen RCTs [39,40,41,42,43,44,45,46,47,48,49,50,51,52] investigated the efficacy of injection treatment for trigger fingers. The therapeutic substances used were corticosteroids and hyaluronic acid. See Table 1 and Table 2.

Injection treatment should be recommended as the first-choice treatment given its effectiveness and low risk of complications. The use of ultrasound guidance, according to Cecen et al. [39], does not provide greater clinical benefits compared to blind injection but increases costs, time, and procedure difficulty.

Hyaluronic acid is less effective than corticosteroid in reducing short-term painful symptoms but exhibits greater effectiveness in the medium to long term and a lower risk of adverse events [40,41].

#### 3.1.1. Injection Technique

All authors used palmar injections. Jiménez et al. [42] compared the effects of corticosteroid infiltration through two different access points and found that dorsal web space access would result in less post-procedure pain with the same clinical benefits as palmar access. The technique described by various studies (Cecen et al. [39], Liu et al. [40], Jiménez et al. [42], Yildirim et al. [43], Tajik et al. [50], and Kosiyatrakul et al. (2018) [52]) involves infiltration at the level of the A1 pulley with a 45° needle inclination. Cecen et al. [39], Liu et al. [40], Yildirim et al. [43], Sato et al. [44], Hansen et al. [45], Atthakomol et al. [49], and Kosiyatrakul et al. (2018) [52] specify that infiltration was performed at the fibrous nodule, inside the tendon sheath, while Patrinely et al. [46] specify that infiltration was performed outside the tendon sheath. Mardani-Kivi et al. (2018) [51] performed the injection at the level of the A1 pulley with a 30–45° needle inclination, comparing intra-sheath and extra-sheath infiltration.

Kosiyatrakul et al. (2021) [47] demonstrated that a subcutaneous single injection digital block with 2 mL of 1% lidocaine is highly effective in reducing pain associated with corticosteroid infiltration for trigger fingers. On the other hand, Patrinely et al. [46] reported that the administration of anesthetic in solution with corticosteroids is more painful than the single administration of corticosteroids.

#### 3.1.2. Needle

Some authors (Cecen et al. [39], Yildirim et al. [43], Jegal et al. [48], and Mardani-Kivi et al. (2018) [51]) used a 26-gauge needle, while in other studies (Liu et al. [40], Kanchanathepsak et al. [41], Jiménez et al. [42], Kosiyatrakul et al. (2021) [47], and Kosiyatrakul et al. (2018) [52]), a 25 G needle was used. Only one study used a 23 G needle (Hansen et al. [45]), and Patrinely et al. [46] used a 27 G needle. Only Tajik et al. [50] used a 29 G needle.

#### 3.1.3. Drugs

*Type of corticosteroids.* The most commonly used corticosteroids were methylprednisolone 40 mg/mL (Cecen et al. [39], Sato et al. [44], Tajik et al. [50], and Mardani-Kivi et al. (2018) [51]), triamcinolone 40 mg/mL (Hansen et al. [45], Patrinely et al. [46], and Jegal et al. [48]) or 10 mg/mL (Liu et al. [40], Kanchanathepsak et al. [41], Atthakomol et al. [49], and Kosiyatrakul et al. (2021) [47]), and betamethasone 6 mg/mL (Jiménez et al. [42] and Yildirim et al. [43]). Kosiyatrakul et al. (2018) [52] used triamcinolone acetonide comparing three different dosages (5 mg, 10 mg, and 20 mg).

*Type of Hyaluronic Acids.* Liu et al. used a moderate molecular weight (1000–12,000 kd), and Kanchanathepsak et al. used a low-molecular-weight hyaluronic acid (500–730 kD, 20 mg/2 mL).

#### 3.1.4. Outcomes

Numerous studies (Cecen et al. [39], Liu et al. [40], Kanchanathepsak et al. [41], Jiménez et al. [42], Yildirim et al. [43], Jegal et al. [48], and Atthakomol et al. [49]) used the VAS scale to quantify pain before and after treatment. Instead, Kosiyatrakul et al. (2021) [47] used the VAS scale during the injection procedure, and Patrinely et al. [46] used the VAS scale immediately after the procedure to identify the least painful technique. Tajik et al. [50] used the NPRS (Numerical Pain Rating Scale) to evaluate pain before and after treatment. Kanchanathepsak et al. [41] and Yildirim et al. [43] used the DASH scale to quantify upper limb functionality and pain. Liu et al. [40] and Sato et al. [44] used TAM analysis (total active motion) to quantify ROM in flexion–extension. Cecen et al. [39], Liu et al. [40], Yildirim et al. [43], Jegal et al. [48], and Mardani-Kivi et al. (2018) [51] used the Quinnell scale to quantify functional improvement. Jegal et al. [48] also used the PGI-I scale (modified patient global impression of improvement). Hansen et al. [45] used the TFS (trigger finger score) and evaluated palpation pain and the incidence of complications. Tajik et al. [50] used a digital dynamometer to assess the grip strength, evaluated the stage of stenosing tenosynovitis (SST), and used the Boston questionnaire for symptom severity and functional status. Liu et al. [40] also used the dynamometer and the MHQ scale (Michigan Hand Outcome Questionnaire). Atthakomol et al. [49] also used the MHQ scale.

#### 3.1.5. Clinical Results

Hansen et al. [45] and Sato et al. [44] compared corticosteroid injection treatment with surgical A1 pulley release (open or percutaneous): surgery has a higher healing rate than corticosteroid injections, although injection allows for fewer complications, less pain, and better ROM in the first months post-procedure. Corticosteroid injections therefore remain a good first-line treatment given the low morbidity associated with the procedure and do not preclude or prevent the patient from undergoing surgery later. In addition, Jegal et al. [48] demonstrated that simultaneous corticosteroid injection with percutaneous release would reduce the perceived pain symptoms in the first months post-procedure.

In the study by Cecen et al. [39], the effectiveness of ultrasound-guided infiltrative therapy versus blind infiltrative therapy was compared for trigger finger. Following infiltration, both groups experienced a statistically significant improvement in terms of pain reduction and Quinnell grading improvement at both follow-ups (6 weeks and 6 months). However, no statistically significant differences were observed between the two groups. Therefore, according to the authors, the use of ultrasound guidance in this case does not provide clinical advantages over blind infiltrative techniques; in fact, it may prolong the procedure duration.

In the study by Mardani-Kivi et al. (2018) [51], the effectiveness of intra-sheath (IS) versus extra-sheath (ES) ultrasound-guided corticosteroid injection was evaluated, considering the functional improvement in the treated finger with the Quinnell grading system (0–4). Before treatment, in both groups, most patients had a Quinnell grade of 2 (54.2% of patients in the ES group and 57.8% of patients in the IS group). Quinnell grade improved statistically significantly for both groups: in fact, at all follow-ups, almost all of the patients in both groups had a Quinnell grade of 0. However, no statistically significant difference has been documented between the ES group and the IS group. Therefore, the intra- and extra-sheath injection of CS plus lidocaine determines the same results in terms of finger function. This result means that it is not necessary to use US-guided injections to be sure that the injection has been performed inside the tendon sheath, with a consequent saving of time, costs, and labor.

Kosiyatrakul et al. (2018) [52] compared three different dosages of triamcinolone acetonide (5 mg, 10 mg, and 20 mg) for the treatment of trigger fingers and thumbs, in patients with grade II, IIIA, or IIIB based on Green classification. The evaluated outcomes were pain and triggering of the affected finger, but the authors did not specify the scales used to evaluate these parameters. After the injection, pain and triggering improved by the first and second weeks in all three groups and were nearly completely resolved at 6 weeks. For pain, in the 5 mg group, complete resolution was documented for 75% of patients at 1 week and 97.5% at 6 weeks; in the 10 mg group, for 72.5% of patients at 1 week and 100% at 6 weeks; and in the 20 mg group, for 75% of patients at 1 week and 100% at 6 weeks. For triggering, in the 5 mg group, complete resolution was documented for 47.5% of patients at 1 week and 87.5% at 6 weeks; in the 10 mg group, for 45% of patients at 1 week and 100% at 6 weeks; and in the 20 mg group, for 35% of patients at 1 week and 95% at 6 weeks. However, there were no statistically significant differences among the groups.

Regarding the endpoint of success rate, a dose-related pattern was found at 3 months, and documented success rates were 60% (5 mg group), 90% (10 mg group), and 100% (20 mg group), with a statistically significant difference between the 5 mg group vs. the 10 mg group and between the 5 mg group vs. the 20 mg group. At 6 months, documented success rates were 20% (5 mg group), 35% (10 mg group), and 62.5% (20 mg group), with a statistically significant difference between the 20 mg group versus both the 5 mg group and the 10 mg group. At 9 months and 12 months, there were no statistically significant differences among the groups. The authors do not recommend the dosage of 5 mg triamcinolone acetonide for patients with multiple risk factors for the recurrence of the trigger finger.

Patients with single-digit involvement had a higher success rate but with no statistical significance. Only two patients complained of skin dryness and discoloration in the area of injection.

Yildirim et al. [43] compared the efficacy of corticosteroid infiltrative therapy with extracorporeal shockwave therapy (ESWT) in patients with a grade 2 trigger finger according to the Quinnell score. At the first follow-up, conducted one month after treatment, both groups showed statistically significant differences compared to pre-treatment values for all considered outcomes (VAS; frequency, severity, and functional impact of triggering; Quick-DASH). However, no statistically significant differences were observed between the two groups. Even at subsequent follow-ups, conducted at 3 and 6 months, no significant differences were found in terms of all the clinical assessment parameters. Therefore, ESWT treatment could be as effective as a corticosteroid injection for improving symptom severity and functional status in patients with grade 2 trigger fingers.

In Jimenez et al.’s study [42], dorsal web space injection was compared to palmar midline injection of steroids in the treatment of trigger digits. In the dorsal approach group, the mean VAS was 3.6 (range 0–10), while in the palmar approach group, it was 5.4, with a statistically significant difference between the two groups. Furthermore, in the dorsal approach group, treatment was effective in 67% of cases, whereas in the palmar approach group, it was effective in 56% of cases. However, this difference was not statistically significant. Therefore, the extra-sheath corticosteroid injection through the dorsal web space is less painful and at least as effective as the palmar midline technique. Thus, it should be considered in the initial treatment of the trigger finger and the trigger thumb.

In the study by Kosiyatrakul et al. (2021) [47], they assessed the efficacy of subcutaneous single-injection digital block (SSIDB) for reducing pain during corticosteroid injection for the trigger finger. Patients were randomized into three groups: SSIDB with 2 mL of 1% lidocaine, SSIDB with 1 mL of 1% lidocaine, and a control group without digital block. The VAS pain scores for corticosteroid needle insertion in both SSIDB groups were significantly lower than those in the control group. Additionally, the VAS pain score during corticosteroid infiltration in the 2 mL group was significantly lower than that in the 1 mL group and the control group. Therefore, a single-injection subcutaneous digital block with 2 mL of 1% lidocaine proved highly effective in reducing pain associated with corticosteroid injection for the trigger finger.

In the study by Patrinely et al. [46], the efficacy of corticosteroid injection with and without local anesthetic was compared in the trigger finger. Immediate post-injection pain scores were significantly higher for injections containing lidocaine with epinephrine compared to the placebo (VAS 3.5 vs. 2.0, on a scale of 0–10). This study highlights that injection-associated pain is greater when lidocaine with epinephrine is combined with corticosteroids. Utilizing corticosteroids alone not only reduces pain but also proves to be simpler, more effective, and safer.

Atthakomol et al. [49] compared the efficacy of splinting, steroid injection, or splinting plus steroid injection in adults with trigger fingers. There were no clinically important differences in VAS pain scores and MHQ scores among the three treatment groups at any follow-up. Therefore, according to this study, splinting is recommended as the initial treatment for the absence of clinically important differences between splinting alone and CS injection alone. The combination of CS injection and splinting is disadvantageous because the benefits in terms of pain reduction and functional improvement are not different from those achieved with CS injection or splinting alone.

Tajik et al. [50] compared the efficacy of steroid injection versus steroid injection plus splinting. For the primary outcome (NPRS), the analysis of baseline versus 1 and 3 months of data showed that both interventions were successful in decreasing the intensity of pain (from 8.3 at baseline to 5.9 at 1 month for the CS group, and from 7.8 at baseline to 4.3 at 1 month in the CS + splint group). However, after 3 months, the improvement was significantly higher for the CS + splint group (3.9 mean value) versus the CS group (6.4 mean value). Both groups experienced an improvement in the secondary outcomes (grip strength, SST stages, and Boston questionnaire) 1 and 3 months after the intervention, with a statistically significant difference between the two groups at 3 months in the symptom severity scale (SSS) of the Boston questionnaire and in the stages of stenosing tenosynovitis (SST).

Two studies compared the efficacy of hyaluronic acid (HA) infiltration versus corticosteroid infiltration: in the study by Liu et al. [40], the type of hyaluronic acid used was not specified, while in the study by Kanchanathepsak et al. [41], low-molecular-weight hyaluronic acid (LMW-HA) (500–730 kD, 20 mg/2 mL) was used. Liu et al. demonstrated that HA has good therapeutic effects comparable to those of corticosteroids. MHQ scores continued to increase in the HA group but not in the corticosteroid group at the 3-month follow-up. Therefore, HA presents several advantages over corticosteroids: it induces a continuous, specific functional improvement in the hand and reduces the risk of adverse events. Kanchanathepsak et al. highlighted that HA and corticosteroids have comparable therapeutic effects in the treatment of trigger fingers. However, corticosteroid injection has a faster effect in reducing pain and inflammation.

**Table 1 jfmk-09-00146-t001:** Summary Tables of Rcts Concerning Trigger Finger Infiltration Therapy Corticosteroid.

Article	Population Sample	Injection Technique	Experimental Group	Control Group	Follow-Up	Outcomes	Conclusions
Sato et al., 2012 [44]	N = 137 Trigger finger grade 2–4 on the Quinnell grading system.	- Injection into the osteofibrous canal. - At the level of the A1 pulley.	CS GROUP: injection of 2 mL of methylprednisolone acetate (40 mg/mL). Possible second infiltration one month later in case of failure of the procedure.	PERCUTANEOUS RELEASE GROUP: unlocking the A1 pulley with a 40 × 12 needle, using longitudinal movements in the direction of the axis of the flexor tendon. OPEN SURGERY GROUP: surgical incision of 2 cm transverse to the axis of the finger at the palmar skin fold, followed by a subcutaneous dissection and the longitudinal opening of the A1 pulley.	1 and 2 weeks, and 1, 2, 4, and 6 months.	Cure (remission of the trigger for 6 months); relapse within 6 months; therapeutic failure with persistence of the trigger after surgical treatment or after the second injection; local pain following the procedure; joint pain at the IP joint of the thumb and at the PIP joint of the fingers; ROM (total active motion—TAM); complications.	Open and percutaneous interventions allow for a higher healing rate than CS infiltrations (100% vs. 57% after one CS infiltration and 86% after two CS infiltrations). On the other hand, infiltration is associated with less pain in the first month after the procedure. With regard to ROM, open surgery causes worse results in the first 4 months.
Cecen et al., 2015 [39]	N = 74	- Infiltration into the synovial sheath. - Just distal to the A1 pulley. - 26 G needle.	ULTRASOUND GUIDED INFILTRATION (Bodor and Flossman method) 40 mg/mL methylprednisolone. In case of failure of the procedure, a second infiltration was performed after one month.	NON-ULTRASOUND-GUIDED INFILTRATION 40 mg/mL methylprednisolone. In case of failure of the procedure, a second infiltration was performed after one month.	6 weeks and 6 months.	Pain (VAS), Quinnell grading system (to evaluate relapses).	CS injection is recommended as the primary (valid, low cost, and low risk) treatment option for the trigger finger. The use of ultrasound guidance increases cost, time, and difficulty without clinical benefits compared to the blind technique.
Yildirim et al., 2016 [43]	N = 40 Trigger finger grade 2 on the Quinnell grading system.	- Infiltration in the palmar side. - 45° needle inclination. - Distally inside the nodule near the A1 pulley. - 26 G needle.	CS GROUP: 0.5 mL betamethasone + 0.5 mL lidocaine 2%. Possible second infiltration after 3 months in case of persistence of symptoms.	ESWT GROUP: 3 sessions per week, 1000 pulses, energy 2.1 bar, frequency 15 Hz.	Baseline and 1, 3, and 6 months.	Pain (VAS), disability (q-DASH), Quinnell grading system (patient considered cured if equal to 0).	CS and ESWT have the same therapeutic efficacy, so ESWT may be a less invasive treatment option, particularly for patients who have contraindications to CS.
Hansen et al., 2017 [45]	N = 165 Trigger finger grade >2 on the Quinnell grading system.	- Ultrasound-guided infiltration. - An injection inside the sheath with 1 mL of corticosteroid, and an injection outside the sheath with 1 mL of corticosteroid. - Close to the pulley A1. - Palm side. - 23 G needle.	CS GROUP: 1 mL triamcinolone acetonide (40 mg/mL) + 1 mL lidocaine (10 mg/mL).	OPEN SURGERY GROUP: surgical incision of 2 cm transverse to the axis of the finger at the palmar skin fold, followed by a subcutaneous dissection and the longitudinal opening of the A1 pulley.	Baseline and 3, and 12 months.	Trigger finger score (TFS), pain upon palpation, complications.	Open surgery is superior to ultrasound-guided CS injections. On the other hand, complications after open surgery are greater; CS injections therefore remain a good first-line treatment due to the low morbidity associated with the procedure. Furthermore, the injection does not preclude or prevent the patient from undergoing surgery later.
Mardani-Kivi et al., 2018 [51]	N = 166	- Ultrasound-guided infiltration. - Palmar side. - 30–45° needle inclination. - At the A1 pulley. - 26 G needle. - 40 mg/mL methylprednisolone acetate (1 mL) + 0.5 mL of 2% lidocaine.	INTRA-SHEATH GROUP: infiltration administered outside the tendon sheath.	EXTRA-SHEATH GROUP: infiltration administered inside the tendon sheath, with US views that confirmed tendon sheath distention.	Baseline, 3 and 6 weeks, and 3, 6, and 12 months.	Quinnell grading system.	No difference between ES and IS injection in terms of efficacy on the finger’s functionality. This conclusion implies the lack of necessity of US-guided injections, which increases cost, time, and labor.
Kosiyatrakul et al., 2018 [52]	N = 120 Trigger finger grade II, IIIA, or IIIB based on Green’s classification.	- Injection into the flexor tendon sheath. - Palmar side. - 45° needle inclination. - At the A1 pulley. - 25 G needle.	5 mg GROUP (A) vs. 10 mg GROUP (B) vs. 20 mg GROUP (C): (A): 0.5 mL of 10 mg/mL triamcinolone acetonide + 0.5 mL of 1% lidocaine. (B): triamcinolone acetonide 10 mg/mL without any dilution. (C): 0.5 mL of 40 mg/mL triamcinolone acetonide + 0.5 mL of 1% lidocaine. All patients received 1 mL of injected fluid. Before the injection, the skin and subcutaneous tissue were anesthetized with 2 mL of 1% lidocaine.		Baseline, 1, 2, and 6 weeks, and 3, 6, 9, and 12 months.	Pain and triggering of the finger.	There is an association between outcome and dosage of CS in the first 6 months. Triamcinolone acetonide 5 mg had a lower success rate during that period of time. Patients with single-digit involvement had a higher success rate. Only 2 patients complained of skin dryness and discoloration in the area of injection.
Jegal et al., 2019 [48]	N = 120 Trigger finger > 6 months unresponsive to conservative treatments (including CS injection).	- Injection through the operative site. - Palm side. - 26 G needle.	PERCUTANEOUS PULLEY RELEASE GROUP: percutaneous release of the A1 pulley with a specially designed hook-shaped knife.	PERCUTANEOUS RELEASE GROUP + CS INFILTRATION: injection of 0.5 mL of triamcinolone (40 mg/mL) at the end of the percutaneous release procedure.	Baseline, 3 weeks, and 3 months.	Pain (VAS), modified patient global impression of improvement (PGI-I), modified Quinnel grade.	Simultaneous steroid injection at the time of surgical release provides greater subjective improvement in the early period after percutaneous trigger finger release.
Jimenez et al., 2020 [42]	N = 160	- Palmar access: injection with the needle inclined at 45° at the level of the metacarpal head. - Dorsal access: injection with a 25 G needle, directing the needle towards the subcutaneous tissue at the level of the head of the metacarpal, in the dorsal radial or dorsal ulnar area depending on the case.	GROUP WITH PALMAR ACCESS: 1 mL of betamethasone 6 mg/mL + 1 mL mepivacaine 2%.	GROUP WITH DORSAL ACCESS: 1 mL of betamethasone 6 mg/mL + 1 mL mepivacaine 2%.	Baseline and 1, 3, and 12 months.	Pain (VAS)	The extra-sheath corticosteroid injection through the dorsal web space is less painful and at least as effective as the palmar midline technique. So, it should be considered in the initial treatment of the trigger finger and trigger thumb.
Kosiyatrakul et al., 2021 [47]	N = 90 Trigger finger grades 1–3 of Green’s classification.	Digital block with anesthetic: - identification of the flexor tendon at the level of the metacarpal head. - injection into the subcutaneous space above the A1 pulley. - 30 G needle. CS infiltration: - injection into the flexor tendon sheath at the head of the metacarpal. - 25 G needle.	Subcutaneous single-injection digital block (SSIDB) with 2 mL of 1% lidocaine, before injecting 1 mL of 10 mg triamcinolone acetonide.	SSIDB with 1 mL of 1% lidocaine, before injecting 1 mL of 10 mg triamcinolone acetonide. CONTROL GROUP: application of an ethyl chloride spray on the skin at the level of the metacarpal head, before injecting 0.5 mL of 1% lidocaine and 0.5 mL of triamcinolone acetonide.	Post-procedure.	Pain during the procedure (VAS).	Single-injection subcutaneous digital block with 2 mL of 1% lidocaine was highly effective in reducing pain associated with CS injection for the trigger finger.
Patrinely et al., 2021 [46]	N = 73	- Injection at the A1 pulley level. - 27 G needle.	CS + ANESTHETIC GROUP: triamcinolone (1 mL, 40 mg) + 1% lidocaine with epinephrine (1 mL).	CS GROUP: triamcinolone (1 mL, 40 mg) + normal saline (1 mL, placebo).	Post-procedure.	Post-procedure pain (VAS with an image of the Wong–Baker Pain FACES Pain Rating Scale).	In treating trigger fingers, corticosteroid injections are effective and have relatively little associated pain. This study demonstrates that injection-associated pain is greater when lidocaine with epinephrine is combined with CS. Using CS is not only less painful but also more simple, effective, and safe.
Tajik et al., 2022 [50]	N = 60	- Injection at the A1 pulley level. - 29 G needle.	CS GROUP: a single injection of 40 mg/mL methylprednisolone + 0.5 mL of lidocaine.	CS GROUP + SPLINT: a single injection of 40 mg/mL methylprednisolone + 0.5 mL of lidocaine. Then, immobilization of the MCP joint with a static thermoplastic splint that blocks the MCP joint in a neutral position (24 h a day).	Baseline and 1 and 3 months.	Pain (NPRS) and functionality with grip strength (dynamometer), stages of stenosing tenosynovitis (SST), Boston questionnaire that includes symptom severity scale (SSS) and functional status scale (FSS).	The full-time wearing of a static MCP splint for 3 months immediately following a single CS + lidocaine injection increases and stabilizes the clinical benefits of CS treatment.
Atthakomol et. al., 2023 [49]	N = 120	- Injection into the flexor tendon sheath.	CS GROUP: 1 mL of 1% lidocaine without epinephrine and 1 mL of triamcinolone acetonide (10 mg/mL). SPLINT + CS GROUP	SPLINT GROUP: involved the patient wearing a fixed metacarpophalangeal joint orthosis in the neutral position at least 8 h per day for 6 consecutive weeks.	6, 12, and 52 weeks after the intervention.	Pain (VAS) and Michigan Hand Outcomes Questionnaire (MHQ).	Splinting alone is recommended as the initial treatment for trigger finger because there were no clinically important differences between splinting and CS injection alone in terms of pain reduction and functional improvement of up to 1 year. The combination of CS injection and splinting is disadvantageous because the benefits in terms of pain reduction and functional improvement are not different from those achieved with CS injection or splinting alone.

**Table 2 jfmk-09-00146-t002:** Hyaluronic acid vs. corticosteroid.

Article	Population Sample	Injection Technique	Experimental Group	Control Group	Follow-Up	Outcomes	Conclusions
Liu et al., 2015 [40]	N = 36 Trigger finger grade ≥ 1 on the Quinnell grading system.	- Ultrasound-guided infiltration. - Palmar access. - At the level of the A1 pulley. - Inside the sheath of the flexor tendons. - 25 G needle. - Needle inclination at 45°.	HA GROUP: 1 mL of HA.	CS GROUP: 1 mL of triamcinolone 10 mg/mL.	Baseline, 3 weeks, and 3 months.	Quinnell trigger finger scale, Michigan Hand Outcome Questionnaire (MHQ) scale, pain (VAS), total active movement (TAM) scale, grip strength (dynamometer).	HA had good therapeutic effects with effects similar to those of CS. MHQ scores continued to increase in the HA group and not in the CS group at the 3-month follow-up. HA therefore presents several advantages compared to CS: it induces a continuous specific functional improvement in the hand and reduces the risk of adverse events.
Kanchanathepsak et al., 2020 [41]	N = 51 Trigger finger grade 1,2, or 3 on the Quinnell grading system with symptoms < 6 months.	- US-guided infiltration. - At the level of the A1 pulley. - From the palmar side. - 25 G needle.	HA GROUP: 1 mL LMW-HA (500–730 kD, 20 mg/2 mL).	CS GROUP: 1 mL di 10 mg/mL triamcinolone.	Baseline and 1, 3, and 6 months.	Pain (VAS) and disability (DASH).	HA and CS have a comparable therapeutic effect in the treatment of trigger fingers. However, CS injection has a faster effect in reducing pain and inflammation.

### 3.2. De Quervain’s Tenosynovitis

Nine randomized controlled clinical trials [53,54,55,56,57,58,59,60,61] investigated the efficacy of the injection treatment in de Quervain’s tenosynovitis. The therapeutic substances used were corticosteroids, NSAIDs, and hyaluronic acid. See Table 3, Table 4 and Table 5.

All authors unanimously agree on its effectiveness. In particular, Hadianfard et al. [53] found that corticosteroid injection is more effective than acupuncture, and Suwannaphisit et al. [54] observed the superiority of corticosteroid injection treatment over NSAID injection in terms of grip strength improvement, pain reduction, and disability.

Moreover, according to Shin et al. [55], performing the procedure under ultrasound guidance is no more effective than blind infiltration. In cases where there is a complete septum in the first extensor compartment, infiltrating only the subcompartment of the short thumb extensor tendon is as effective as infiltrating both subcompartments with a lower risk of complications, as it involves a lower dose of corticosteroids [56].

Das et al. [61] compared the efficacy of thumb abduction orthosis versus corticosteroid injection and concluded that the difference between the two therapeutic options is small in a long-term follow-up.

Three authors investigated using a brace after the injection. According to Akhtar et al. [57], the combination of steroid and brace is more effective than using the thumb brace alone. Mardani-Kivi et al. (2014) [58] found that the combination of corticosteroid injection and brace use is more effective than corticosteroid alone; however, according to Ippolito et al. [59], there are no significant differences between the combination of steroid and brace compared to corticosteroid infiltrative treatment alone.

Finally, Orlandi et al. [60] investigated the efficacy of hyaluronic acid injection treatment combined with corticosteroid and concluded that the addition of low-molecular-weight hyaluronic acid improves outcomes in terms of pain and disability.

#### 3.2.1. Injection Technique

Shin et al. [55] and Orlandi et al. [60] used the volar approach, while Mardani-Kivi et al. (2014) [58] and Ippolito et al. [59] chose the dorsal approach. Hadianfard et al.’s study [53] and Das et al.’s study [61] specified the needle inclination angle, respectively, of 30–45 degrees and 45 degrees. Numerous studies (Suwannaphisit et al. [54], Mardani-Kivi et al. (2014) [58], and Ippolito et al. [59]) injected at the point of maximum tenderness to digital pressure, while in the study of Hadianfard et al. [53] and Das et al. [61], the injection was made 1 cm proximally to the radial styloid process. In most of the considered studies (Shin et al. [55], Jung et al. [56], and Orlandi et al. [60]), the injection was performed with the needle positioned perpendicular to the major axis of the tendons in the first compartment; Hadianfard et al. [53] and Das et al. [61] injected parallel to the course of the tendons. Regarding the use of ultrasound guidance, Jung et al. [56] and Orlandi et al. [60] used an in-plane approach with a transverse probe to the major axis of the tendons.

#### 3.2.2. Needle

In most studies (Hadianfard et al. [53], Shin et al. [55], Jung et al. [56], and Ippolito et al. [59]), a 25 G needle was used; Akhtar et al. [57] used 24 or 26 G needles. Orlandi et al. [60] used a 22 G needle.

#### 3.2.3. Drugs

Type of corticosteroids. Several studies (Hadianfard et al. [53], Mardani-Kivi et al. (2014) [58], Ippolito et al. [59], Orlandi et al. [60], and Das et al. [61]) used methylprednisolone acetate 40 mg/mL. Suwannaphisit et al. [54] and Akhtar et al. [57] used triamcinolone acetonide 10 mg/mL; Shin et al. [55] and Jung et al. [56] used triamcinolone acetonide 40 mg/mL.

Type of Hyaluronic Acid. Orlandi et al. used a low-molecular-weight hyaluronic acid (0.8%, 16 mg/2 mL).

#### 3.2.4. Outcomes

For pain assessment, all studies used the VAS scale (Hadianfard et al. [53], Shin et al. [55], Jung et al. [56], Akhtar et al. [57], Mardani-Kivi et al. (2014) [58], Ippolito et al. [59], Orlandi et al. [60], and Das et al. [61]), except Suwannaphisit et al. [54], who used the verbal-NRS scale. For disability assessment, Ippolito et al. [59] used the DASH questionnaire; Suwannaphisit et al. [54] used the Thai-DASH questionnaire; Hadianfard et al. [53], Akhtar et al. [57], Mardani-Kivi et al. (2014) [58], Orlandi et al. [60], and Das et al. [61] used the quick-DASH questionnaire. Only Shin et al.’s study [55] considered the PRWE scale (patient-rated wrist evaluation), and only Ippolito et al. [59] included the Finkelstein test. Suwannaphisit et al. [54] also used a dynamometer to assess grip strength. Finally, Orlandi et al. [60] performed an ultrasound evaluation and considered the patients’ subjective outcomes.

#### 3.2.5. Clinical Results

In the study by Hadianfard et al. [53], the effectiveness of corticosteroid injection and acupuncture was compared. The mean pre-treatment Q-DASH score for all patients was 62.8. In patients treated with corticosteroid injection, the mean Q-DASH score (0–100) decreased from 61.2 to 13.7 after two weeks of treatment and to 6.1 after six weeks of treatment. In patients treated with acupuncture, the mean Q-DASH score decreased from 64.4 to 24.3 and 9.8 (at two and six weeks post-treatment, respectively). The mean pre-treatment VAS score (0–10) for all patients was 6.9. In patients treated with corticosteroid infiltration, the mean VAS score decreased from 6.67 to 2.53 after two weeks of treatment and to 1.20 after six weeks of treatment. In patients treated with acupuncture, the mean VAS score decreased from 7.13 to 3.9 and 2.07 (at two and six weeks post-treatment, respectively). Therefore, both corticosteroid infiltration and acupuncture represent two effective therapeutic options in the treatment of de Quervain’s tenosynovitis, with superiority of infiltration.

In the study by Das et al. [61], the effectiveness of thumb abduction orthosis was compared to CS infiltration, considering pain (VAS) and functionality (Q-DASH) parameters. There was a significant improvement in both mean VAS and Q-DASH scores at 1, 3, and 6 months in both groups. In particular, at the 1-month follow-up, the CS group significantly improved compared to the orthosis group, in both VAS and Q-DASH scores: the orthosis group went from 6.8 mean VAS at baseline to 3.3, while the CS group went from 6.9 to 2.0; the orthosis group went from 59.07 mean Q-DASH at baseline to 23.71, while the CS group went from 59.31 to 8.63. However, there were no statistically significant differences between the two groups at 3- and 6-month follow-ups.

In the study by Mardani-Kivi et al. (2014) [58], the effectiveness of CS infiltration combined with a thumb spica cast (TSC) was compared to CS infiltration alone, considering functionality (Q-DASH) and pain (VAS) parameters. At the first follow-up (3 weeks post-treatment), the CS + TSC group showed a therapeutic success rate of 97%, while the CS-only group showed 76%. At the second follow-up (6 months post-treatment), the CS + TSC group demonstrated a therapeutic success rate of 93%, while the CS-only group showed 69%. Both treatment approaches showed improvement in terms of functionality and pain, but the combined CS + TSC approach had a significantly higher treatment success rate.

In the study by Akhtar et al. [57], the effectiveness of CS infiltration with casting was compared to casting alone, considering functionality (Q-DASH) and pain (VAS) parameters. At the follow-up conducted 6 weeks post-treatment, the combined treatment with infiltration and casting demonstrated clear superiority with a therapeutic success rate of 85.1%, compared to casting alone with a success rate of 37.5%.

In the study by Shin et al. [55], the effectiveness of ultrasonography-guided and blind CS injection was compared, considering pain and functionality parameters. Pre-treatment VAS values (6.8 for the ultrasound-guided group and 6.5 for the blind group) decreased to 1.6 and 2.1, respectively, at the 4-week follow-up, and further reduced to 1.4 and 1.5 at the 3-month follow-up. Pre-treatment PRWE values (54.1 for the ultrasound-guided group and 61.3 for the blind group) decreased to 16.5 and 21.6, respectively, at the 4-week follow-up, and further reduced to 13.9 and 17.3 at the 3-month follow-up.

In the study by Ippolito et al. [59], the effectiveness of CS infiltration alone was compared to the combined therapy of CS infiltration plus immobilization with a brace for 3 weeks. At the 6-month follow-up, a significant improvement in both considered outcomes (DASH and VAS) was demonstrated in both groups. Additionally, at 6 months, in the CS group, 88% of patients reported the resolution of at least two out of three clinically present symptoms before treatment (radial wrist pain, tenderness to palpation, and positive Finkelstein test), and in the CS + brace group, this occurred in 73% of patients. Therefore, both groups experienced clinical improvement, but without a statistically significant difference. Finally, at the 6-month follow-up, in the group treated with infiltration alone, resolution of radial wrist pain was observed in 100% of patients, compared to 63% of patients who received combined treatment.

In the study by Jung et al. [56], the effectiveness of CS infiltration in the two subcompartments (APL and EPB tendons) was compared to infiltration in the EPB compartment alone. The mean VAS values significantly decreased from pre-treatment (mean VAS of 67 for both groups) to the follow-up conducted at 6 weeks after treatment (mean VAS of 10 for both groups); at the 12-week follow-up, the mean VAS was 12 for the APL + EPB group and 11 for the EPB group. Therefore, the infiltration of only one subcompartment (EPB) is as effective as the infiltration of both subcompartments. Furthermore, a targeted injection into the EPB subcompartment alone may reduce the steroid dose used, potentially decreasing complications.

In the study by Suwannaphisit et al. [54], the effectiveness of NSAID (ketorolac) infiltration was compared to CS infiltration. At the 6-week follow-up, patients in the CS group had a statistically lower average pain score (VAS, 0–10) compared to the NSAID group (0.7 vs. 5.3), a higher DASH functional score (4.4 vs. 34.1), higher right grip strength (60.8 vs. 49.2), and higher left grip strength (59.8 vs. 50.3). However, there was no difference in pinch strength. Complete resolution of symptoms at 6 months occurred in 90% of cases with CS compared to 40% of cases with NSAID.

In the study by Orlandi et al. [60], the effectiveness of ultrasound-guided infiltration with CS alone (group A); CS followed by a saline solution after 15 days (group B); and CS followed by low-molecular-weight hyaluronic acid (LMW-HA) after 15 days (group C) was compared. At baseline, the mean tendon thickness measured ultrasonographically in the three groups was 1.6 mm (group CS), 1.4 mm (group CS + SF), and 1.7 mm (group CS + LMW-HA). At the 3-month follow-up, the mean values measured in the three groups were 0.7 mm, 0.8 mm, and 0.5 mm, respectively. At 6 months post-treatment, the mean values measured were 1.5 mm, 1 mm, and 0.7 mm, respectively. At baseline, the mean VAS score was 6 for all three treatment groups. At the 1-month follow-up, the mean VAS score decreased to 2 for all three treatment groups. At the 3-month follow-up, the mean values were 3 for the CS group and 1 for both the CS + SF and CS + LMW-HA groups. At the 6-month follow-up, the mean values were 3 for the CS group, 2 for the CS + SF group, and 1 for the CS + LMW-HA group. At baseline, the mean qDASH score was 55 for the CS group, 56 for the CS + SF group, and 55 for the CS + LMW-HA group. At the 1-month follow-up, the mean qDASH score decreased to 23, 22, and 21, respectively. At the 3-month follow-up, the mean values in the three groups were 27, 25, and 23. At the 6-month follow-up, the mean value was 51 for both the CS and CS + SF groups and 26 for the CS + LMW-HA group.

Therefore, at the 6-month follow-up, patients who received the combined treatment CS + LMW-HA showed a statistically significant improvement in all three considered outcomes compared to baseline values.

### 3.3. Potential Biases in the Review Process

Despite the methodical search, it is possible that some eligible studies could be missed. The searches resulted in a large number of results, due to the sensible search string. We utilized adjunctive filters such as systematic reviews and humans. In determining study eligibility, review authors had to decide if the rehabilitation provided between intervention and control groups was the same, except for the amount of time spent. Infiltrative injections are a complex intervention, which naturally varies from patient to patient. Moreover, the technique of the physician could influence the results. The overall risk of bias in the included studies is shown in Figure 2 and Figure 3.

**Table 3 jfmk-09-00146-t003:** Summary tables of RCTs concerning infiltration therapy of De Quervain’s Tenosynovitis.

Article	Sample	Injection Technique	Experimental Group	Control Group	Follow-Up	Outcomes	Conclusions
Hadianfard et al., 2013 [53]	N = 30	- Infiltration 1 cm proximal to the radial styloid process. - Needle angled 30–45° from the skin. - The direction of the needle parallel to the direction of the tendons of the first extensor compartment. - 25 G needle.	CS GROUP: 1 infiltration with 1 mL methylprednisolone acetate + 1 mL lidocaine 2%.	ACUPUNCTURE GROUP: 5 sessions of 30 min.	Baseline and 2 and 6 weeks.	Disability (Q-DASH), pain (VAS).	Both treatments are effective, but CS has greater efficacy.
Mardani-Kivi et al., 2014 [58]	N = 67	- Infiltration in the first dorsal compartment in the area of maximum tenderness upon palpation.	CS + BRACE GROUP: 1 infiltration of 40 mg methylprednisolone acetate + 1 cc of lidocaine 2% + fiberglass brace for 3 weeks.	CS GROUP: 1 infiltration 40 mg of methylprednisolone acetate + 1 cc of lidocaine 2%.	Baseline, 3 weeks, and 6 months.	Disability (Q-DASH), pain (VAS).	The CS + brace association is more effective than CS alone.
Akhtar et al., 2020 [57]	N = 134	- Infiltration into the first extensor compartment. - 24 or 26 G needle.	CS + BRACE GROUP: 1 infiltration of 1 mL (10 mg) of triamcinolone acetonide and 1 mL of 1% lidocaine hydrochloride + thumb splint.	BRACE GROUP	Baseline and 2 and 6 weeks.	Disability (Q-DASH), pain (VAS).	The CS + brace association is more effective than the thumb brace alone.
Shin et al., 2020 [55]	N = 48	- Infiltration into the first extensor compartment. - 0.5 mL triamcinolone acetate 40 mg/mL -25 G needle. - The blind injection was performed with a volar approach perpendicular to the axis of the tendons of the first compartment.	US-GUIDED INJECTION GROUP	BLIND INJECTION GROUP	Baseline, 4 weeks, and 3 months.	Pain and disability (VAS, patient-rated wrist evaluation—PRWE).	CS gives good clinical results with no statistically significant difference between US-guided and blinded procedures.
Ippolito et al., 2020 [59]	N = 26	- Infiltration in the first dorsal compartment in the area of maximum tenderness upon palpation. - 25 G needle.	CS GROUP: 1 infiltration of 1 mL 40 mg of methylprednisolone acetate + 2 mL 2% lidocaine.	CS GROUP + BRACE for 3 weeks (can only be removed for bathing)	Baseline, 3 weeks, and 6 months	Disability (DASH), pain (VAS), Finkelstein test.	Both treatments are effective without statistically significant differences, but at 6 months, the brace could compromise the resolution of the radial side wrist pain.
Jung et al., 2021 [56]	N = 48	- US-guided infiltration (in-plane short axis) in the first compartment of the extensor tendons. - 25 G needle. - 0.5 mL of 2% lidocaine + 0.5 mL triamcinolone acetonide (40 mg/mL). TOTAL 1 mL	INFILTRATION GROUP OF THE 2 SUBCOMPARTMENTS: infiltration into subcompartments of the APL and EPB tendons. 0.5 mL in APL 0.5 mL in EPB	INFILTRATION GROUP OF 1 SUBCOMPARTMENT: infiltration of the EPB only. 0.5 mL in EPB	Baseline, 6 weeks, and 3 months.	Pain (VAS), complications.	Infiltration of only one subcompartment (EPB) in patients with de Quervain syndrome with a complete septum is as effective as infiltration of both subcompartments. Furthermore, a targeted injection into the EPB subcompartment alone may reduce the steroid dose used, potentially decreasing complications.
Das et al., 2021 [61]	N = 60	- Infiltration 1 cm proximal from the tip of the styloid process of the radius, into the first extensor compartment. - Needle angled 45° from the skin, parallel to the direction of the tendons.	BRACE GROUP: thermoplastic thumb abduction orthosis that keeps the wrist in a neutral position, and the thumb at 90° abduction and 15–20° extension. 20 h a day for one month.	CS GROUP: 1 infiltration of 1 mL (40 mg) of methylprednisolone acetate.	Baseline and 1, 3, and 6 months.	Disability (Q-DASH), pain (VAS).	CS has a higher improvement rate in patient symptoms, but the difference between CS and brace is small in long-term follow-ups. The brace is useful in patients at risk for side effects of CS use.

**Table 4 jfmk-09-00146-t004:** CS vs. NSAIDs.

Article	Sample	Injection Technique	Experimental Group	Control Group	Follow-Up	Outcomes	Conclusions
Suwannaphisit et al., 2022 [54]	N = 64	1 infiltration along the longitudinal axis of the tendons of the first extensor compartment, either just proximal or distal to the radial styloid, at the point of maximum pain.	CS GROUP: 1 mL triamcinolone acetonide 10 mg/mL + 0.5 mL 1% xylocaine with adrenaline.	NSAID GROUP: 1 mL ketorolac 30 mg/mL + 0.5 mL 1% lidocaine with adrenaline.	Baseline and 6 weeks.	Disability (Thai-DASH), pain (verbal-NRS), grip strength (dynamometer).	Triamcinolone performs better in all outcomes: - Complete resolution of symptoms at 6 months: triamcinolone in 90% of cases, ketorolac in 40% of cases. - DASH scores: on average, lower values (indicating better functionality) with triamcinolone. -Improved grip strength recovery with triamcinolone.

**Table 5 jfmk-09-00146-t005:** CS vs. CS + HYALURONIC ACID.

Article	Sample	Injection Technique	Experimental Group	Control Group	Follow-Up	Outcomes	Conclusions
Orlandi et al., 2014 [60]	N = 75	Ultrasound-guided infiltration of the 1st extensor compartment of the wrist. - A volar in-plane approach with the probe positioned transversally to the tendons of the first extensor compartment. - 22 G needle.	CS + HA GROUP: 1 infiltration of 1 mL methylprednisolone acetate + 1 infiltration after 15 days with LMW-HA (0.8%, 16 mg/2 mL).	CS GROUP: 1 infiltration with 1 mL methylprednisolone acetate 40 mg/mL. CS + SF GROUP: 1 infiltration of 1 mL methylprednisolone acetate + 1 infiltration after 15 days of 2 mL of 0.9% saline solution.	Baseline (US, VAS, q DASH), after 1 month (VAS, qDASH), and after 3 and 6 months (US, VAS, qDASH).	Pain (VAS), disability (qDASH), US evaluation, patient’s subjective outcome.	LMW-HA infiltration after CS infiltration improves all the outcomes (US, clinical, and subjective).

## 4. Discussion

This systematic review investigated the efficacy of infiltrative therapy in the treatment of pain in hand tendinopathies, in particular, the clinical efficacy of infiltrative therapy in trigger finger and de Quervain’s tenosynovitis was evaluated.

According to the results from this systematic review, in the treatment of the trigger finger, all the authors recommended infiltrative treatment as the first therapeutic choice to reduce pain and to improve functionality. The use of hyaluronic acid has demonstrated greater effectiveness in the medium/long term together with a lower risk of adverse events, compared to corticosteroids. The most commonly used infiltrative technique involves injection at the level of the A1 pulley, at the site of the fibrous nodule and within the tendon sheath, with the needle inclined at 45°. The majority of studies used a 25 G needle. Among corticosteroids, the most used was triamcinolone 40 mg/mL. One study compared the effectiveness of infiltrative therapy and ESWT therapy, concluding that the latter can constitute an effective alternative in patients with contraindications to the use of corticosteroids or those who do not wish to undergo infiltrative therapy. From the comparison between the infiltrative approach and the surgical A1 pulley release (open or percutaneous), it emerged that surgical treatment determines a higher cure rate, but infiltrative therapy is associated with a lower rate of complications, less pain, and better ROM; therefore, it should be considered as the first therapeutic option.

The infiltrative therapy of de Quervain’s tenosynovitis is essentially based on the use of corticosteroids and all authors agree in defining the local injection of these drugs as effective in the treatment of this pathology, resulting in a reduction in pain and an improvement in functionality. The most used substance is methylprednisolone acetate 40 mg/mL. In addition, a study demonstrated that the therapeutic association of corticosteroids and LMW-HA determines an improvement in pain and function outcomes compared to the use of corticosteroids alone. The infiltrative technique can be performed according to a dorsal or volar approach, generally at the level of the point of maximum pain upon palpation. The use of a 22 G needle is preferred. The needle is directed perpendicular to the major axis of the tendons of the first compartment and inclined at 30–45°. The use of ultrasound guidance is recommended using an in-plane technique with a probe perpendicular to the major axis of the tendons.

## 5. Conclusions

Infiltrative therapy of the trigger finger and de Quervain’s tenosynovitis constitutes a fundamental element in the treatment of these pathological conditions. In fact, the injection of therapeutic substances such as corticosteroids and hyaluronic acid into the peritendinous area allows for excellent results to be achieved in terms of pain reduction and improvement in the functionality of the hand. Further RCTs are needed to evaluate the efficacy of infiltrative therapy in the other hand tendinopathies.

## Figures and Tables

**Figure 1 jfmk-09-00146-f001:**
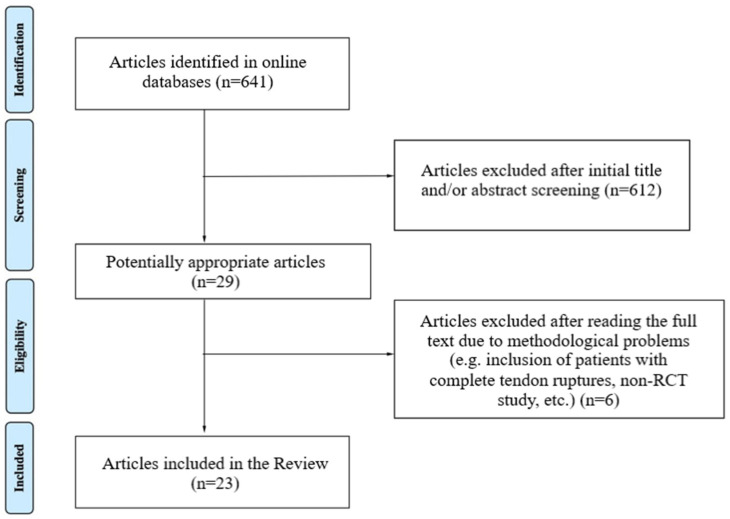
PRISMA flowchart.

**Figure 2 jfmk-09-00146-f002:**
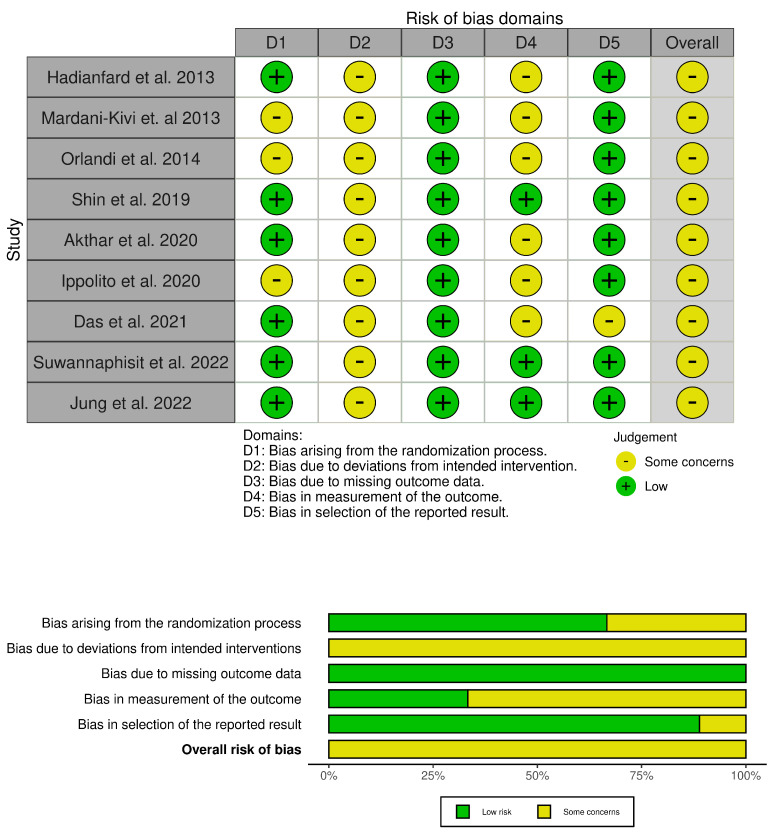
RoB of de Quervain’s tenosynovitis articles [53,54,55,56,57,58,59,60,61].

**Figure 3 jfmk-09-00146-f003:**
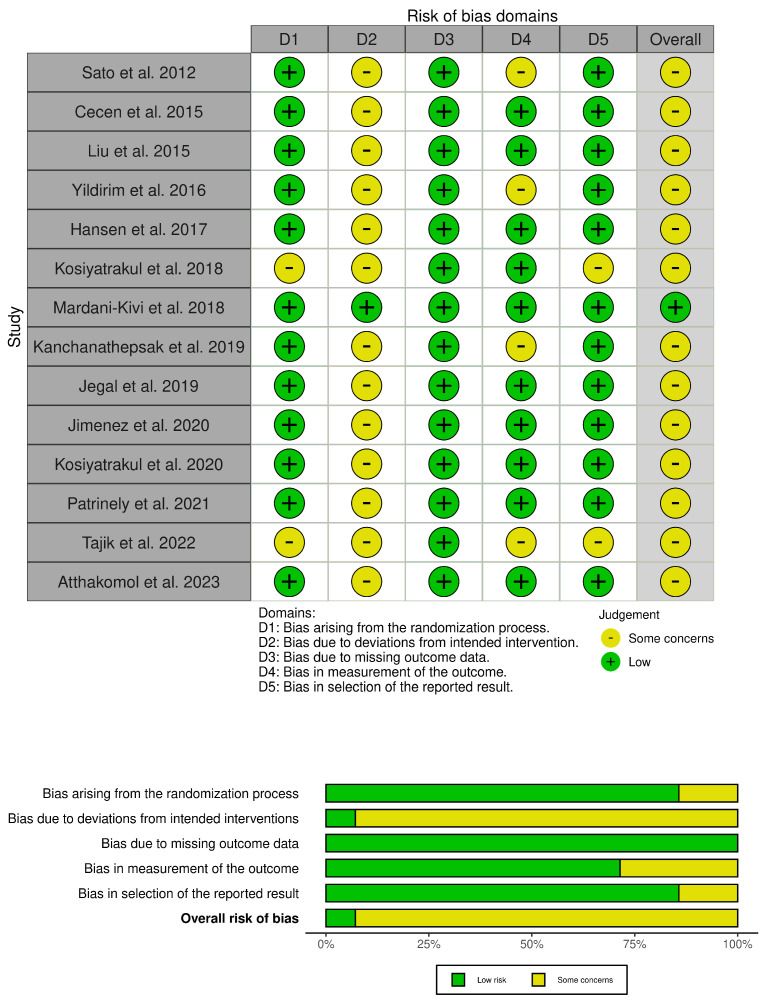
RoB of trigger finger articles [39,40,41,42,43,44,45,46,47,48,49,50,51,52].

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
