# Peer review of "A Practical Guide to Injection Therapy in Hand Tendinopathies: A Systematic Review of Randomized Controlled Trials"

_jfmk, 2024, doi:10.3390/jfmk9030146_

Round 1

Reviewer 1 Report

Comments and Suggestions for Authors

I am a search strategy specialist. My review pertains only to the search strategy part of the manuscript.

A. Choice of databases

The consulted databases are sufficient and well chosen.

B. Search strategy - I have the following remarks concering the search strategy.

The search as given by the authors retrieves 146 references when the search is limited to the last 10 years, limited to the english language, and limited to the publication type Randomized controlled trial. As the exact details of the search strategy are missing, it is unclear if this publication type has been used as a filter for RCTs. Applying the RCT filter as applied by e.g. the Cochrane Library, the number increases to 291 references. This is quite a large difference compared to the complete set as reported (88).

Adding several alternative terms, the results in PubMed increase from 291 to 409 references. It would appear that potential relevant articles have been missed.

Search as given by the authors:

((injection) OR (infiltrative treatment) OR (infiltration)) AND ((tendinopathy) OR (tendinopathies) OR (tendinosis) OR (tendon injury) OR (tendon inflammation) OR (tendon degeneration)) AND ((upper limb) OR (wrist) OR (thumb) OR (hand))

Authors search with added limits (last 10 years, RCTs):

((injection OR infiltrative treatment OR infiltration) AND (tendinopathy OR tendinopathies OR tendinosis OR tendon injury OR tendon inflammation OR tendon degeneration) AND (upper limb OR wrist OR thumb OR hand)) AND ("2014/01/01"[PDAT] : "3000/12/31"[PDAT]) AND ("clinical trial"[pt] OR "clinical trial"[tiab] OR "clinical trials as topic"[mesh] OR "clinical trials"[tiab] OR "control groups"[mesh] OR "control group"[tiab] OR "control groups"[tiab] OR "controlled clinical trial"[pt] OR "controlled clinical trials as topic"[mesh] OR "cross-over studies"[mesh] OR "cross over study"[tiab] OR "cross over studies"[tiab] OR "double-blind method"[mesh] OR "double blind"[tiab] OR "evaluation studies as topic"[mesh] OR "follow-up studies"[mesh] OR "follow up study"[tiab] OR "follow up studies"[tiab] OR "placebos"[mesh] OR placebo*[tiab] OR placebos*[tiab] OR "pragmatic clinical trial"[pt] OR "prospective studies"[mesh] OR "prospective study"[tiab] OR "prospective studies"[tiab] OR "RaCT"[tiab] OR "RaCTs"[tiab] OR "random allocation"[mesh] OR "randomised "[tiab] OR "randomized controlled trial"[pt] OR "randomized controlled trials as topic"[mesh] OR "randomized"[tiab] OR random*[tiab] OR "RCT"[tiab] OR "RCTs"[tiab] OR "Research Design"[MeSH:noexp] OR "Research design"[tiab] OR "Research designs"[tiab] OR "single blind"[tiab] OR "single-blind method"[mesh] OR ((single*[tiab] OR double*[tiab] OR triple*[tiab]) AND (blind*[tiab] OR mask*[tiab])) OR volunteer*[tiab] OR "trial"[ti] OR "trials"[ti])

Search with added terms by reviewer:

(("Injections"[Mesh] OR Injections OR injection OR inject* OR microinjections OR microinjection OR microinject* OR "infiltrative treatment" OR "infiltrative treatments" OR "infiltrative therapy" OR "infiltrative therapies" OR infiltration OR "Hyaluronic Acid"[Mesh] OR "Hyaluronic Acid" OR "Adrenal Cortex Hormones"[Mesh] OR "corticosteroids" OR "corticosteroid" OR "corticosteroid*" OR "Platelet-Rich Plasma"[Mesh] OR "platelet-rich plasma" OR "PRP" OR "autologous blood" OR "autologous cell" OR "Sclerotherapy"[Mesh] OR "Sclerotherap*" OR "sclerotherapy" OR "Prolotherapy"[Mesh] OR "prolotherapy" OR "Botulinum Toxins"[Mesh] OR "botulinum toxin" OR "botulinum toxins") AND ("Tendinopathy"[Mesh] OR tendinopathy OR tendinopathies OR tendinosis OR Tendinitis OR Tendonitis OR "tendon injury" OR "tendon injuries" OR "tendon inflammation" OR "tendon degeneration" OR "tendon injury"[title/abstract:~6] OR "tendon injuries"[title/abstract:~6] OR "tendon inflammation"[title/abstract:~6] OR "tendon degeneration"[title/abstract:~6] OR "trigger finger" OR "de quervain's" OR "de quervains" OR "de quervain" OR "dequervain*") AND ("Upper Extremity"[Mesh] OR "Upper Extremity" OR "Upper Extremities" OR "upper limb" OR "upper limbs" OR wrist OR thumb OR hand OR wrists OR thumbs OR hands)) AND ("2014/01/01"[PDAT] : "3000/12/31"[PDAT]) AND ("clinical trial"[pt] OR "clinical trial"[tiab] OR "clinical trials as topic"[mesh] OR "clinical trials"[tiab] OR "control groups"[mesh] OR "control group"[tiab] OR "control groups"[tiab] OR "controlled clinical trial"[pt] OR "controlled clinical trials as topic"[mesh] OR "cross-over studies"[mesh] OR "cross over study"[tiab] OR "cross over studies"[tiab] OR "double-blind method"[mesh] OR "double blind"[tiab] OR "evaluation studies as topic"[mesh] OR "follow-up studies"[mesh] OR "follow up study"[tiab] OR "follow up studies"[tiab] OR "placebos"[mesh] OR placebo*[tiab] OR placebos*[tiab] OR "pragmatic clinical trial"[pt] OR "prospective studies"[mesh] OR "prospective study"[tiab] OR "prospective studies"[tiab] OR "RaCT"[tiab] OR "RaCTs"[tiab] OR "random allocation"[mesh] OR "randomised "[tiab] OR "randomized controlled trial"[pt] OR "randomized controlled trials as topic"[mesh] OR "randomized"[tiab] OR random*[tiab] OR "RCT"[tiab] OR "RCTs"[tiab] OR "Research Design"[MeSH:noexp] OR "Research design"[tiab] OR "Research designs"[tiab] OR "single blind"[tiab] OR "single-blind method"[mesh] OR ((single*[tiab] OR double*[tiab] OR triple*[tiab]) AND (blind*[tiab] OR mask*[tiab])) OR volunteer*[tiab] OR "trial"[ti] OR "trials"[ti])

C. Other

Please use the plural of "randomized controlled trial" in the title.

Reviewer 2 Report

Comments and Suggestions for Authors

This is an interesting comprehensive review of the use and clinical efficacy of infiltrative therapies for hand tendinopathy. I believe that some changes could be made to improve the final version of the manuscript.

1. The introductory section should be shortened and limited to mentioning hyaluronan and corticosteroid therapy.

2. You should present the risk bias analysis for each group of tendinopathy evaluated. You should also include a weighted bar chart and discuss more about the risk of bias of your systematic review.

3. Please review the results described in the text to avoid redundant information in the tables. 

4. Please replace "cortisone" with "corticosteroid" throughout the manuscript.

Comments on the Quality of English Language

Minor issues detected.

Round 2

Reviewer 1 Report

Comments and Suggestions for Authors

The search date should be modified.

An appendix with the reproducible search strategies should be added.

Comments on the Quality of English Language

No comment

Reviewer 2 Report

Comments and Suggestions for Authors

All my concerns were addressed

Author Response

All my concerns were addressed.

Thank you for the very positive comment.